# OpenReview forum: "LiveNewsBench: Evaluating LLM Web Search Capabilities with Freshly Curated News"
_ICLR.cc/2026/Conference — Submitted to ICLR 2026_

### Official Review · Reviewer_HD95 · 2025-10-30

**Soundness:** 3
**Presentation:** 3
**Contribution:** 2
**Rating:** 2
**Confidence:** 3

**Summary:**

This paper proposes LiveNewsBench, a benchmark designed to evaluate the web research capabilities of LLMs. It uses an automatic pipeline to collect recent news articles and then prompts LLMs to generate QA pairs based on them. The authors also commit to maintaining the benchmark for two years and evaluate a wide suite of systems.

**Strengths:**

* The pipeline is straightforward and easily scalable.
* The experiments are comprehensive, testing a wide variety of both closed-source and open-source models across different sizes.
* The authors include a human-verified test set (200 questions) and show that the discrepancy between results on unverified and verified sets is minimal.

**Weaknesses:**

* The LLM-based quality control appears weak. As stated in the paper, during human annotation, around 20% of the QA pairs that passed automated verification were rejected by annotators. This means that the training, validation, and most of the test sets (except for the human-verified subset) contain roughly 20% noise, which is quite high. The paper should explore ways to further reduce this noise. Although this does not seem to heavily affect evaluation results (Section 4.2), it could pose issues when training on these noisy data.
* The test set size is relatively small. For example, SimpleQA contains 4K QA pairs, while LiveNewsBench’s test set includes only 800 questions, and the Longitudinal Q&A Set just 300 questions.
* The benchmark requires ongoing human maintenance. What happens after two years when it is no longer maintained? The long-term impact of the benchmark could be questioned because of this.
* The paper lacks discussion on the copyright and licensing issues of using news articles to construct datasets. Some websites may have stricter content licenses than others.

**Questions:**

* What is the total cost of generating the entire dataset?
* The dataset partitioning seems unusual. The paper states,
> In the current release, the training set includes over 6,800 Q&A pairs based on news events from January to May 2025. The validation set contains 680 samples from June 2025, and the test set comprises 800 samples from July and August 2025.

Why are the training, validation, and test sets split chronologically rather than randomly? Wouldn’t random assignment yield a more representative evaluation?

---

> ### Author Response · Authors · 2025-11-21
> **Response to Reviewer HD9530**
>
> > Weakness 1: The LLM-based quality control appears weak. As stated in the paper, during human annotation, around 20% of the QA pairs that passed automated verification were rejected by annotators. This means that the training, validation, and most of the test sets (except for the human-verified subset) contain roughly 20% noise, which is quite high. The paper should explore ways to further reduce this noise. Although this does not seem to heavily affect evaluation results (Section 4.2), it could pose issues when training on these noisy data.
>
> Thank you for raising this concern. We would like to clarify that our human verification process is primarily conducted to enhance reliability and out of an abundance of caution. The ~20% of rejected questions may be slightly ambiguous or have answers that could change over time. However, they are often still answerable via search and do not equate to noise. As a result, human verification does not significantly affect the overall benchmark results. As discussed in Section 4.2 (Page 7, 344-358) and Table 2 (page 6, 290-302, reproduced below), both the accuracy and the relative ordering of models on the full test set remain stable with those on the human-verified subset. This suggests that our fully automated datasets can reliably measure LLM search capabilities.
>
> Additionally, in response to the feedback during the rebuttal, we are committed to further improving our QA quality evaluation via human annotation. We have already initiated a human evaluation study, which includes assessments of QA quality and measurements of inter-annotator agreement. We will report the results of this additional validation and update the paper PDF accordingly once the study is complete.
>
> | Model     | Human-verified accuracy (%) | Full test accuracy (%) | Delta | Rank change (Human-verified → Full) |
> |--------|-------|-----|-----:|-------------|
> | GPT-5     | 91.0    | 87.9                   |    -3.1    | 1 → 1                      |
> | DeepSeek V3.1 Thinking (685B) | 83.0                        | 84.4                   |    +1.4    | 3 → 2                      |
> | Claude Sonnet 4             | 83.5                        | 81.9                   |    -1.6    | 2 → 3                      |
> | Qwen3 235B A22B Instruct 2507 | 76.0                        | 76.2                   |    +0.2    | 4 → 4                      |
> | Gemini 2.5 Pro                | 75.5                        | 75.6                   |    +0.1    | 5 → 5                      |
>
> > Weakness 2: The test set size is relatively small. For example, SimpleQA contains 4K QA pairs, while LiveNewsBench’s test set includes only 800 questions, and the Longitudinal Q&A Set just 300 questions.
>
> Thank you for highlighting this concern. Due to high inference costs, modern LLM benchmarks are often limited in size. For instance, GPQA Diamond (STEM reasoning) contains fewer than 200 samples, SWE-Bench Verified (Software Engineering Agents) has 500, and Seal-0, a search benchmark we cite, includes only 111 questions. In comparison, our human-verified test set includes 200 questions, and our full test set comprises 800, which aligns with these precedents.
>
> Moreover, evaluating agentic tasks further amplifies token usage, as each action requires resending the full message history to the model API, leading to double billing. In practice, evaluating a single model on 200 samples can consume 1-2 million tokens. Given the need to test multiple models and ablations, our choice of test set size reflects a practical trade-off between cost and evaluation robustness. Since our Q&A pair creation process is largely automated, and both the code and dataset are fully open-sourced via a URL provided in the paper (page 2, line 102), users can easily generate a larger dataset themselves if needed.
>
> > Weakness 3: The benchmark requires ongoing human maintenance. What happens after two years when it is no longer maintained? The long-term impact of the benchmark could be questioned because of this.
>
> The two-year update commitment specified in the paper represents a minimum duration. We anticipate continuing updates beyond this period depending on community interest and adoption, particularly for the training, validation, and test sets, which do not require human supervision. Additionally, both our code and data are fully open-sourced, enabling the community and any interested users to generate new datasets and maintain the benchmark beyond the initial two-year window.
>
> > Weakness 4: The paper lacks discussion on the copyright and licensing issues of using news articles to construct datasets. Some websites may have stricter content licenses than others.
>
> Thank you for your suggestion. We will include a discussion on the copyright and licensing in the camera-ready version of the paper. We will also explore other methods to address potential copy-right issues, such as redacting the article content from our released dataset and only retaining the article URL.

---

> ### Author Response · Authors · 2025-11-21
> **Response to Reviewer HD9530, Continued**
>
> > Question 1: What is the total cost of generating the entire dataset?
>
> Thank you for your questions. We estimate that the total cost of constructing the dataset is approximately \textdollar 620, or \textdollar 76.5 per 1K Q&A pairs.
>
> > Question 2: The dataset partitioning seems unusual. Why are the training, validation, and test sets split chronologically rather than randomly? Wouldn’t random assignment yield a more representative evaluation?
>
> We chose to partition the dataset chronologically, with the test set comprising the most recent examples, to maximize the time gap between the model's training data cutoff and the events in the test set. This approach minimizes the risk of data contamination and ensures that our benchmark more accurately reflects the model’s ability to search for information online, instead of recalling facts from internal knowledge.
>
> Please let us know if you have any further questions or concerns. We are happy to address any additional issues you may have. Thank you again for your thoughtful feedback.

---

### Official Review · Reviewer_ri14 · 2025-11-01

**Soundness:** 2
**Presentation:** 3
**Contribution:** 2
**Rating:** 4
**Confidence:** 5

**Summary:**

The paper introduces LiveNewsBench, a scalable and regularly updated benchmark for evaluating the web search capabilities of large language models (LLMs). The authors argue that existing benchmarks fail to distinguish between retrieval and memorized knowledge because they rely on static question sets that models may already have seen during training. To overcome this, they construct LiveNewsBench from freshly published news articles, automatically generating question–answer pairs that require real-time retrieval and multi-step reasoning. The benchmark includes training, validation, and test sets, as well as a human-verified subset and a longitudinal component that tracks how answers evolve over time. Evaluations show that LiveNewsBench effectively differentiates models' search abilities: leading proprietary models such as GPT-5 and Grok 4 perform best, while smaller open models lag behind. Disabling internet access causes large performance drops, which confirms that the dataset minimizes contamination from pretraining data. The authors conclude that LiveNewsBench provides a reliable, contamination-limited foundation for measuring and improving LLMs' capacity to reason over dynamic, real-world information.

**Strengths:**

- The paper identifies and mitigates a major flaw in existing web-search benchmarks by using recent news events that occur after model training cutoffs, which ensures minimal contamination from pretraining data

- The benchmark's fully automated pipeline produces large quantities of question-answer pairs with limited human input while maintaining reasonable quality

- The benchmark is designed to refresh quarterly, which helps maintain relevance and tests models on genuinely new information

- The results demonstrate clear differentiation among proprietary, open-source, and smaller models, which confirms the benchmark's ability to measure genuine differences in search performance

**Weaknesses:**

- The paper overstates the limitations of static benchmarks. Static questions remain informative when their correct answers have changed since a model's training cutoff, as they still require retrieval rather than recall. The authors have not convincingly shown that they need to keep generating new benchmarks every few months -- a well-designed static benchmark with time-sensitive questions could already achieve the same goal.

- The benchmark conflates retrieval ability with reasoning accuracy. A model may successfully retrieve the correct information yet fail due to flawed reasoning, so the reported performance does not isolate retrieval skill from broader reasoning capability.

- The human-verified subset cannot guarantee the reliability of the full dataset. Since about 20% of automatically generated items are rejected during manual review, a similar error rate may persist in the unverified portion, which undermines claims of overall data quality.

- The reported accuracies of 85-91% suggest that the benchmark questions may not be sufficiently difficult. The small remaining headroom could also reflect label or data errors rather than genuine task difficulty. If top models already perform near ceiling, the benchmark may fail to provide meaningful differentiation or headroom for future progress.

- The benchmark is planned for quarterly updates over only two years, which raises questions about its long-term value.

**Questions:**

The changing question set makes it difficult to track progress or compare results over time with confidence. How can we tell whether higher scores in later rounds reflect true model improvement or just easier questions?

---

> ### Author Response · Authors · 2025-11-21
> **Response to Reviewer ri14**
>
> Thank you for taking the time to review our submission. We deeply appreciate your insightful and constructive feedback. We would like to address the concerns raised:
>
> > Weakness 1: The paper overstates the limitations of static benchmarks. Static questions remain informative when their correct answers have changed since a model's training cutoff, as they still require retrieval rather than recall. The authors have not convincingly shown that they need to keep generating new benchmarks every few months -- a well-designed static benchmark with time-sensitive questions could already achieve the same goal.
>
> Thank you for raising this concern. We argue that a static benchmark with a fixed set of questions but evolving answers, such as FreshQA, may not adequately evaluate the true search capabilities of LLMs. Because the questions remain fixed, they are often grounded in past events whose correct answers do not actually change, even when the benchmark is designed to feature time-sensitive questions. As a result, SotA LLMs may correctly answer these questions using only their internal knowledge about the events, without searching online at all.
>
> To illustrate this, we conducted **additional experiments evaluating SotA LLMs on the FreshQA** in an offline setting, on both the full dataset and the fast-changing subset. Our results confirm that current SotA LLMs perform well on FreshQA without search or Internet access, even on the fast-changing subset. Most notably, **Grok 4 achieves 74.6% accuracy on the full FreshQA dataset and 60.3% on the fast-changing subset without any internet access.** These results demonstrate that FreshQA can already be largely solved by existing models without using web search, and a regularly refreshed and factual QA benchmark is necessary for evaluating the search capabilities of LLMs.
>
>
> | Model                               | Accuracy on LiveNewsBench without Internet | FreshQA Overall Accuracy | FreshQA Fast-Changing Subset Accuracy |
> |-------------------------------------|-----------------------------|------------------|------------------------|
> | Grok 4                              | 23.5%                       | 74.6%            | 60.3%                  |
> | Qwen3 235B A22B Instruct 2507       | 12.5%                       | 65.4%            | 48.1%                  |
> | Qwen3 235B A22B Thinking 2507       | 14.5%                       | 68.0%            | 45.8%                  |
> | GPT-4.1                             | 18.5%                       | 68.0%            | 45.0%                  |
> | DeepSeek V3.1                       | 16.0%                       | 65.2%            | 42.0%                  |
> | GPT-5                             | 25.0%                       | 72.4%            | 40.5%                  |
> | Claude Sonnet 4                    | 12.5%                       | 65.4%            | 38.9%                  |
> | DeepSeek V3.1 Thinking              | 17.5%                       | 66.2%            | 38.2%                  |
>
> > Weakness 2: The benchmark conflates retrieval ability with reasoning accuracy. A model may successfully retrieve the correct information yet fail due to flawed reasoning, so the reported performance does not isolate retrieval skill from broader reasoning capability.
>
> Thank you for raising this concern. We would like to confirm that our web search agent is implemented using a reflect-before-act approach, similar to ReAct [1]. Since reasoning also contributes to more effective search planning and keyword selection, this paradigm has become the default in many agentic LLM search systems. This includes both commercial and open-source web search agent implementations (e.g., OpenAI Deep Research [2], Claude Web Search [3], Open Deep Research [4], Kimi Researcher [5]). Therefore, we believe that including reasoning, as opposed to explicitly disallowing it, is a fair and realistic way to evaluate the agentic web search capabilities of LLMs.
>
> Moreover, many state-of-the-art models, such as GPT-5, Grok 4, and Gemini 2.5 Pro, always perform chain-of-thought (CoT) reasoning before producing responses. It is not possible to disable such CoT reasoning through API or prompting, and the content of the CoT is not returned to the user. As such, for any benchmark aiming to evaluate these models, it is inherently impossible to isolate the contribution of reasoning from other capabilities required for the task.

---

> ### Author Response · Authors · 2025-11-21
> **Response to Reviewer ri14, Continued**
>
> > Weakness 3: The human-verified subset cannot guarantee the reliability of the full dataset. Since about 20% of automatically generated items are rejected during manual review, a similar error rate may persist in the unverified portion, which undermines claims of overall data quality.
>
> > Weakness 4: The reported accuracies of 85-91% suggest that the benchmark questions may not be sufficiently difficult. The small remaining headroom could also reflect label or data errors rather than genuine task difficulty. If top models already perform near ceiling, the benchmark may fail to provide meaningful differentiation or headroom for future progress.
>
> Thank you for raising this concern. We would like to clarify that our human review process is primarily conducted to enhance reliability and out of an abundance of caution. The ~20% of rejected questions may be slightly ambiguous or have answers that could change over time. However, they are often still answerable via search and do not significantly affect the overall benchmark results. As discussed in Section 4.2 (Page 7, 344-358) and Table 2 (page 6, 290-302, reproduced below), both the accuracy and the relative ordering of models on the full test set remain consistent with those on the human-verified subset, even for leading models that are well performing near the ceiling (80-90% accuracy) This suggests that our fully automated test set can reliably measure of LLM search capabilities, and the remaining headroom is still correlated with model capability.
>
> Additionally, in response to the feedback during the rebuttal, we are committed to further improving our QA quality evaluation via human annotation. We have already initiated a human evaluation study, which includes assessments of QA quality and measurements of inter-annotator agreement. We will report the results of this additional validation and update the paper PDF accordingly once the study is complete.
>
> | Model                         | Human-verified accuracy (%) | Full test accuracy (%) | Delta | Rank change (Human → Full) |
> |-------------------------------|-----------------------------|------------------------|-----------:|----------------------------|
> | GPT-5*                        | 91.0                        | 87.9                   |    -3.1    | 1 → 1                      |
> | DeepSeek V3.1 Thinking (685B) | 83.0                        | 84.4                   |    +1.4    | 3 → 2                      |
> | Claude Sonnet 4*              | 83.5                        | 81.9                   |    -1.6    | 2 → 3                      |
> | Qwen3 235B A22B Instruct 2507 | 76.0                        | 76.2                   |    +0.2    | 4 → 4                      |
> | Gemini 2.5 Pro                | 75.5                        | 75.6                   |    +0.1    | 5 → 5
>
> > Weakness 4: The benchmark is planned for quarterly updates over only two years, which raises questions about its long-term value.
>
> The two-year update commitment specified in the paper represents a minimum duration. We anticipate continuing updates beyond this period depending on community interest and adoption, particularly for the training, validation, and test sets, which do not require human supervision. Additionally, both our code and data are fully open-sourced, enabling the community and any interested users to generate new datasets and maintain the benchmark beyond the initial two-year window.
>
> > Question 1: The changing question set makes it difficult to track progress or compare results over time with confidence. How can we tell whether higher scores in later rounds reflect true model improvement or just easier questions?
>
> We would like to clarify that LiveNewsBench evaluation results are intended to be compared within the same release, not across different releases, as each release contains a distinct set of questions. This design is consistent with other regularly updated LLM benchmarks, such as LiveBench[6] and LiveCodeBench[7]. In these benchmarks, models evaluated in previous releases serve as reference points for assessing newly released models within the context of each new benchmark version.
>
> Please let us know if you have any further questions or concerns. We are happy to address any additional issues you may have. Thank you again for your thoughtful feedback.
>
> [1]: ReAct: Synergizing Reasoning and Acting in Language Models
>
> [2]: Introducing deep research, OpenAI Blog
>
> [3]: How we built our multi-agent research system, Engineering at Anthropic Blog
>
> [4]: langchain-ai/open_deep_research, GitHub Repository
>
> [5]: Kimi-Researcher: End-to-End RL Training for Emerging Agentic Capabilities, Moonshot AI Blog
>
> [6]: LiveBench: A Challenging, Contamination-Limited LLM Benchmark
>
> [7]: LiveCodeBench: Holistic and Contamination Free Evaluation of Large Language Models for Code

---

### Official Review · Reviewer_ByQq · 2025-11-01

**Soundness:** 3
**Presentation:** 3
**Contribution:** 2
**Rating:** 2
**Confidence:** 5

**Summary:**

This paper presents LiveNewsBench, a new benchmark focusing on fresh knowledge from Wikipedia events, sourced from relevant news. The benchmark is constructed from a pipeline of retrieving news articles from Wikipedia events archive, matching the events to specific news articles, generating QA pairs with GPT-5, verifying generated answers with Claude Sonnet 4, and human verification of 200 examples. Experiments on the proposed benchmark with different LLMs using search implemented with a ReAct prompt are conducted, demonstrating the usage of the benchmark on fresh news knowledge.

**Strengths:**

- The paper is well written and easy to understand.

- Benchmark construction steps are clearly documented. QA verification with both different models and humans are conducted, making sure the QA qualities are controlled.

- Experiments with the search enabled LLMs demonstrate the usefulness of the proposed benchmark.

**Weaknesses:**

- The novelty of the work is limited, given the prior research in this direction, such as Fresh QA [1] and Daily Oracle [2].

- The news fetching pipeline is indirect and could accumulate errors. The paper takes a backward approach, going from Wikipedia events to find possible url to the actual news articles. Errors could compound in these matching processes. Why not directly fetch the news articles to generate the QA pairs?

- The benchmark construction relies on heavy usage of different LLMs, and many such as GPT-4.1, GPT-5, Claude Sonnet 4 are proprietary. It remains unclear whether the dependency on the SOTA LLMs is critical, and how sensitive the benchmark is to the choices of the models. Also, the versions and capabilities of the proprietary models could change, and it is unclear how to maintain the consistency of the benchmark.

- Following the above reason, the cost of the benchmark construction is not revealed, as well as details of the cost-performance tradeoff.

- The human validation is done by one of the authors, which may possess intrinsic bias in the human study of the research. Human validation guidelines are not demonstrated. Also, human validation shows 20% of rejection rate, which makes it unclear how to parse the significance accuracy shown in Table 2, where scores are between 70% and 90%. Error analysis is lacking, to understand the real quality of the generated QA pairs.

- The benchmark still needs human involvement and curation, per the promised quarterly updates for the next two years. This may limit the practical usage and maintenance of the benchmark.

- The comparison with prior work, such as FreshQA [1] and Daily Oracle [2], lacks details to fairly trust the results to justify the contribution of this work. FreshQA has knowledge of different update frequencies, so it does not make sense to compare to the part of the dataset with mostly static knowledge. Daily Oracle has multiple choice questions, which may bias towards higher accuracy in the offline setting.

- The paper positions the proposed benchmark for evaluating LLM search capabilities. There are two steps, search and reasoning based on the searched results. It is unclear what certain LLMs are lacking; a model not answering a question correctly could be due to the searched results not correct, or the model not able to reason effectively. The experiments and analysis may be sufficient to fully understand the bottleneck for different models.


> [1] (2023) FreshLLMs: Refreshing Large Language Models with Search Engine Augmentation

> [2] (2024) Are LLMs Prescient? A Continuous Evaluation using Daily News as the Oracle

**Questions:**

1. Why not directly use the news articles to extract events and QAs?

2. The questions generated, as demonstrated in Figure 2, may still not be uniquely identifiable, as there are no time markers. Similar events could happen at multiple times.

---

> ### Author Response · Authors · 2025-11-21
> **Response to Reviewer ByQq**
>
> Thank you for taking the time to review our submission. We deeply appreciate your insightful and constructive feedback. We would like to address the concerns raised:
>
> > Weakness 1: The novelty of the work is limited, given the prior research in this direction, such as Fresh QA [1] and Daily Oracle [2].
>
> > Weakness 7: The comparison with prior work, such as FreshQA [1] and Daily Oracle [2], lacks details to fairly trust the results to justify the contribution of this work. FreshQA has knowledge of different update frequencies, so it does not make sense to compare to the part of the dataset with mostly static knowledge. Daily Oracle has multiple choice questions, which may bias towards higher accuracy in the offline setting.
>
> Thank you for your questions regarding the comparison between our work and existing time-sensitive QA benchmarks, such as FreshQA and Daily Oracle. As noted on Page 2, Lines 73-102, LiveNewsBench is designed to rigorously evaluate LLM's agentic web search capabilities. LiveNewsBench differs from previous time-sensitive factual QA benchmarks in several important ways:
>
> 1. LiveNewsBench uses regularly refreshed question-answer pairs that require knowledge after the LLMs’ training data cutoffs. These questions are intentionally complex so the answers could not be easily derived from common sense and therefore requires searching online. This design distinguishes between a model’s memorized knowledge and its actual web search capabilities.
>
> 2. The benchmark features challenging questions that typically require multiple steps of search, browsing, and reasoning, allowing for the evaluation of LLMs' agentic search abilities.
>
> 3. Unlike FreshQA, which rely on manually crafted Q&A pairs, LiveNewsBench includes an automated pipeline for data acquisition and Q&A generation. This allows for scalable dataset construction and includes a training set. For additional reliability, each benchmark release also includes a human-verified test set.
>
> We would like to provide an **extended comparison with FreshQA**. LiveNewsBench features regularly-refreshed question-answer pairs that are grounded in recent news events. In contrast, **FreshQA features fixed questions that are not grounded in recent events**, and only the answers would change over time. The static nature and relatively simple design of FreshQA questions allow SotA LLMs to answer a large percentage of questions correctly via memorization  (Page 9, Lines 470-476). This limits FreshQA's effectiveness in evaluating the true search capabilities of LLMs. Furthermore, all FreshQA questions are manually crafted rather than automatically generated, which limits the scalability of the approach and results in the absence of a training set.
>
> To further illustrate the difference, we conducted additional experiments evaluating SotA LLMs on the FreshQA benchmark in an offline setting, on both the full dataset and the fast-changing subset. Our results confirm that current SotA LLMs perform well on FreshQA without search or Internet access, even on the fast-changing subset. Most notably, **Grok 4 achieves 74.6% accuracy on the full FreshQA dataset and 60.3% on the fast-changing subset without any internet access.** These results demonstrate that FreshQA can already be largely solved by existing models without using web search, and a regularly refreshed and factual QA benchmark is necessary for evaluating the search capabilities of LLMs.
>
>
> | Model                               | Accuracy on LiveNewsBench without Internet | FreshQA Overall Accuracy | FreshQA Fast-Changing Subset Accuracy |
> |-------------------------------------|-----------------------------|------------------|------------------------|
> | Grok 4                              | 23.5%                       | 74.6%            | 60.3%                  |
> | Qwen3 235B A22B Instruct 2507       | 12.5%                       | 65.4%            | 48.1%                  |
> | Qwen3 235B A22B Thinking 2507       | 14.5%                       | 68.0%            | 45.8%                  |
> | GPT-4.1                             | 18.5%                       | 68.0%            | 45.0%                  |
> | DeepSeek V3.1                       | 16.0%                       | 65.2%            | 42.0%                  |
> | GPT-5                             | 25.0%                       | 72.4%            | 40.5%                  |
> | Claude Sonnet 4                    | 12.5%                       | 65.4%            | 38.9%                  |
> | DeepSeek V3.1 Thinking              | 17.5%                       | 66.2%            | 38.2%                  |

---

> ### Author Response · Authors · 2025-11-21
> **Response to Reviewer ByQq, Continued**
>
> Similarly, **compared to Daily Oracle**, LLMs exhibit significantly lower accuracy on LiveNewsBench when evaluated without Internet access. Notably, the offline answering accuracy on Daily Oracle significantly exceeds 25% across all evaluated models. The top-performing model in Daily Oracle offline evaluation, GPT-5, achieved 77%, more than three times the expected accuracy from random guessing (25%). These results suggest that the high offline performance on Daily Oracle is likely due to model memorization rather than guessing the correct  multiple-choice. Consequently, this benchmark may not be suitable for evaluating the search capabilities of LLMs as the majority of the questions could be solved without search.
>
> | Model       | Accuracy on LiveNewsBench without Internet | Daily Oracle Accuracy (offline) |
> |---------|--------------------------------------------|---------------|
> | GPT-5                               | 25.0%                                      | 77.0%                           |
> | Grok 4                              | 23.5%                                      | 69.5% |
> | DeepSeek V3.1 Thinking              | 17.5%                                      | 68.2%                           |
> | GPT-4.1                             | 18.5%                                      | 67.5%                        |
> | Claude Sonnet 4                     | 12.5%                                      | 50.7%                        |
>
> > Weakness 2: The news fetching pipeline is indirect and could accumulate errors. The paper takes a backward approach, going from Wikipedia events to find possible url to the actual news articles. Errors could compound in these matching processes. Why not directly fetch the news articles to generate the QA pairs?
>
> > Question 1: Why not directly use the news articles to extract events and QAs?
>
> We source events from the Wikipedia Current Events page because we want the benchmark to focus on major, globally relevant events with widespread media coverage. Since we extract QA pairs from online articles, we must block the referenced article and news site during evaluation to prevent data leakage. This makes it essential to ensure that each event is widely reported, so LLMs can still retrieve the correct answer by searching for alternative sources. Wikipedia provides a manually curated list of such impactful events, making it a reliable starting point for identifying events with sufficient coverage diversity.
>
> > Weakness 3: The benchmark construction relies on heavy usage of different LLMs, and many such as GPT-4.1, GPT-5, Claude Sonnet 4 are proprietary. It remains unclear whether the dependency on the SOTA LLMs is critical, and how sensitive the benchmark is to the choices of the models. Also, the versions and capabilities of the proprietary models could change, and it is unclear how to maintain the consistency of the benchmark.
>
> We would like to clarify that LiveNewsBench evaluation results are intended to be compared within the same release, not across different releases, as each release contains a distinct set of questions. This design is consistent with other regularly updated LLM benchmarks, such as LiveBench [1] and LiveCodeBench [2]. Within a given release, all Q&A pairs are generated and evaluated using the same set of LLMs, ensuring that evaluation results are comparable.
>
> In response to the feedback received during the rebuttal, we are currently working on implementing a version of LiveNewsBench using open-source LLMs, and will compare its evaluation results against the standard version built with proprietary models. This process requires re-running the benchmark and will therefore take additional time. We will update the comments and the paper accordingly once this is completed. Furthermore, while open-weight LLMs generally lag behind proprietary ones in performance, their capabilities are rapidly improving. As such, we expect that a version of LiveNewsBench built on open models will soon approach the quality of the version based on proprietary models available today.
>
> > Weakness 4: Following the above reason, the cost of the benchmark construction is not revealed, as well as details of the cost-performance tradeoff.
>
> Thank you for your questions. We estimate that the total cost of constructing the dataset is approximately \textdollar 620, or \textdollar 76.5 per 1K Q&A pairs.

---

> ### Author Response · Authors · 2025-11-21
> **Response to Reviewer ByQq, Continued**
>
> > Weakness 5: The human validation is done by one of the authors, which may possess intrinsic bias in the human study of the research. Human validation guidelines are not demonstrated. Also, human validation shows 20% of rejection rate, which makes it unclear how to parse the significance accuracy shown in Table 2, where scores are between 70% and 90%. Error analysis is lacking, to understand the real quality of the generated QA pairs.
>
> Thank you for raising this concern. We would like to clarify that our human review process is primarily conducted to enhance reliability and out of an abundance of caution. The ~20% of rejected questions may be slightly ambiguous or have answers that could change over time. However, they are often still answerable via search and do not significantly affect the overall benchmark results. As discussed in Section 4.2 (Page 7, 344-358) and Table 2 (page 6, 290-302, reproduced below), both the accuracy and the relative ordering of models on the full test set remain consistent with those on the human-verified subset, even for leading models that are well performing near the ceiling (80-90% accuracy) This suggests that our fully automated test set can reliably measure of LLM search capabilities, and the remaining headroom is still correlated with model capability.
>
> Additionally, in response to the feedback during the rebuttal, we are committed to further improving our QA quality evaluation via human annotation. We have already initiated a human evaluation study, which includes assessments of QA quality and measurements of inter-annotator agreement. We will report the results of this additional validation and update the paper PDF accordingly once the study is complete.
>
> > Weakness 6: The benchmark still needs human involvement and curation, per the promised quarterly updates for the next two years. This may limit the practical usage and maintenance of the benchmark.
>
> Thank you for your feedback. The two-year update commitment specified in the paper represents a minimum duration. We anticipate continuing updates beyond this period depending on community interest and adoption, particularly for the training, validation, and test sets, which do not require human supervision. Additionally, both our code and data are fully open-sourced, enabling the community and any interested users to generate new datasets and maintain the benchmark beyond the initial two-year window.
>
> > Weakness 8: The paper positions the proposed benchmark for evaluating LLM search capabilities. There are two steps, search and reasoning based on the searched results. It is unclear what certain LLMs are lacking; a model not answering a question correctly could be due to the searched results not correct, or the model not able to reason effectively. The experiments and analysis may be sufficient to fully understand the bottleneck for different models.
>
> Thank you for raising this concern. We would like to confirm that our web search agent is implemented using a reflect-before-act approach, similar to ReAct [3]. Since reasoning also contributes to more effective search planning and keyword selection, this paradigm has become the default in many agentic LLM search systems. This includes both commercial and open-source web search agent implementations (e.g., OpenAI Deep Research [4], Claude Web Search [5], Open Deep Research [6], Kimi Researcher [7]). Therefore, we believe that including reasoning, as opposed to explicitly disallowing it, is a fair and realistic way to evaluate the agentic web search capabilities of LLMs.
>
> Moreover, many state-of-the-art models, such as GPT-5, Grok 4, and Gemini 2.5 Pro, always perform chain-of-thought (CoT) reasoning before producing responses. It is not possible to disable such CoT reasoning through API or prompting, and the content of the CoT is not returned to the user. As such, for any benchmark aiming to evaluate these models, it is inherently impossible to isolate the contribution of reasoning from other capabilities required for the task.

---

> ### Author Response · Authors · 2025-11-21
> **Response to Reviewer ByQq, Continued**
>
> > Question 2: The questions generated, as demonstrated in Figure 2, may still not be uniquely identifiable, as there are no time markers. Similar events could happen at multiple times.
>
> We would like to clarify that Figure 2 includes two examples from the human-verified test set and one from the longitudinal Q&A set. Only the training, validation, test, and human-verified Test Sets are designed to contain uniquely identifiable and temporally stable questions. We believe the specific events, locations, and additional constraints in the two samples from the human-verified test set ensure sufficient uniqueness. The longitudinal Q&A set is not intended for training or benchmarking purposes, as it does not guarantee stable answers.
>
> Please let us know if you have any further questions or concerns. We are happy to address any additional issues you may have. Thank you again for your thoughtful feedback.
>
> [1]: LiveBench: A Challenging, Contamination-Limited LLM Benchmark
>
> [2]: LiveCodeBench: Holistic and Contamination Free Evaluation of Large Language Models for Code
>
> [3]: ReAct: Synergizing Reasoning and Acting in Language Models
>
> [4]: Introducing deep research, OpenAI Blog
>
> [5]: How we built our multi-agent research system, Engineering at Anthropic Blog
>
> [6]: langchain-ai/open_deep_research, GitHub Repository
>
> [7]: Kimi-Researcher: End-to-End RL Training for Emerging Agentic Capabilities, Moonshot AI Blog

---

### Official Review · Reviewer_F4vC · 2025-11-01

**Soundness:** 3
**Presentation:** 4
**Contribution:** 2
**Rating:** 4
**Confidence:** 5

**Summary:**

The paper introduces a regularly (quarterly) updated news question answering benchmark, to evaluate the capability of LLMs to answer questions about events beyond their training data by using search and reasoning.  The benchmark is generated in a mostly automatic way, by collecting news stories and prompting LLMs (GPT-4.1, GPT-5) to generate and validate question-answer pairs based on the news.  Some of the questions require a mix of search and reasoning (for example, computing the sum of multiple numbers found via search).  A portion of the test data is human-validated and evaluation is performed via LLM as a judge.  The paper demonstrates that current models perform quite well on the benchmark when given access to search (up to 90+%, with many models getting accuracies in the 80-90% range), but fairly poorly without search (15-25% accuracy).

My current assessment of this paper is a marginal reject.  The need for such a benchmark is clear, and the approach is reasonable.  However, I am not convinced that the work makes a sufficient research contribution beyond prior work on regularly updated news QA benchmarks, and I have a few other concerns about details of the execution (see weaknesses below).

**Strengths:**

- The motivation for the benchmark is clear:  There is a need to better evaluate search-enabled models on events beyond their training.

- The approach of combining question generation, validation, and human verification seems likely to produce quality QA pairs (though at the moment this can't be completely validated -- see weaknesses below).

- The paper is generally well-written.

**Weaknesses:**

**1**)  It is not clear to me that the paper makes a sufficient research contribution beyond prior work on "live" QA benchmarks.  Because it requires human validation, the benchmark cannot be updated arbitrarily frequently.  The authors commit to updating it quarterly for 2 years.  This seems similar to prior efforts, such as FreshQA and RealTimeQA.  The paper states that RealTimeQA is not "continuously updated"; but if my understanding is correct, it was continuously updated (weekly) for some time until it wasn't -- just like the authors of this submission promise to do.  It is not obvious to me how different the proposed benchmark is from extending RealTimeQA.  In the case of FreshQA, the paper states that FreshQA is too easy for LLMs without search, but I am unclear about this argument (see weakness #2).  I do see some other differences between these benchmarks (e.g., RealTimeQA had a much smaller number of questions), but I don't know whether these differences are sufficient to produce different conclusions when models are tested on them.  (Here I am ignoring the comparison the paper makes with Daily Oracle, since the task there is future prediction, which is different from factual QA.)

**2**)  Regarding the comparison with FreshQA:  The paper states that GPT-5 obtains an accuracy of 72.4% on it.  The FreshQA paper however lists accuracies of ~30-45% for GPT-4 (depending on the choice of "strict" or "relaxed" metric -- which is being used here?) and <15% for fast-changing questions, which are the most comparable to LiveNewsBench.  Is GPT-5 that much better than GPT-4 without search?  Could the authors confirm that they reproduce the accuracy of GPT-4 on FreshQA listed in the FreshQA paper?  How does GPT-5 do on the fast-changing subset of FreshQA?  These questions seem important to address in order to make a convincing case for the need for LiveNewsBench over FreshQA.

**3**)  The human validation is done by an author of the paper, and the paper states that the human validator rejects about 20% of the automatically generated QA pairs.  Since the authors could be biased, an additional validation should be done by independent human validators to get an unbiased estimate of QA quality and ideally to measure inter-human agreement as well.

**4**)  The benchmark questions test both search and reasoning capabilities.  This can be seen as a strength, but on the other hand, the fact that these skills are mixed in a single question means that we cannot separate a model's basic reasoning ability (such as mathematical skills) from its ability to search for new knowledge.  Since reasoning ability has nothing to do with the "live" nature of the benchmark, the case for including it is not clear to me.  Perhaps it would make sense to at least separate the questions into multiple classes, depending on whether they are expected to require search, reasoning, or both?

**5**)  A more minor weakness:  The paper states that the benchmark will be updated on a quarterly basis.  This may be sufficient for many purposes, but it does mean that when a new model comes out, it may take a full quarter before it can be meaningfully evaluated.

**6**)  Also minor:  The benchmark relies on GPT-4.1 and GPT-5 for question generation and evaluation.  These are closed models whose availability and features are out of the authors' control, which could introduce instability into the benchmark updates.

**7**)  The provided URL, livenewsbench.com, does not appear to be live.

**Questions:**

- I have a hard time following the longitudinal example in Fig. 2.  What makes it longitudinal?
- Table 3 doesn't appear to be referenced anywhere.  I assume it should have been referenced in Sec. 4.3?

---

> ### Author Response · Authors · 2025-11-21
> **Response to Reviewer F4vC**
>
> Thank you for taking the time to review our submission. We deeply appreciate your insightful and constructive feedback. We would like to address the concerns raised:
>
> >  Weakness 1: It is not clear to me that the paper makes a sufficient research contribution beyond prior work on "live" QA benchmarks. This seems similar to prior efforts, such as FreshQA and RealTimeQA.
>
> > Weakness 2: Regarding the comparison with FreshQA: The paper states that GPT-5 obtains an accuracy of 72.4% on it. … Is GPT-5 that much better than GPT-4 without search? the authors confirm that they reproduce the accuracy of GPT-4 on FreshQA listed in the FreshQA paper? How does GPT-5 do on the fast-changing subset of FreshQA?
>
> Thank you for your questions regarding the comparison between our work and existing time-sensitive QA benchmarks, such as FreshQA and RealtimeQA. As noted on Page 2, Lines 73-102, LiveNewsBench is designed to rigorously evaluate LLM's agentic web search capabilities. LiveNewsBench differs from previous time-sensitive factual QA benchmarks in several important ways:
>
> 1. LiveNewsBench uses regularly refreshed question-answer pairs that require knowledge after the LLMs’ training data cutoffs. This design distinguishes between a model’s memorized knowledge and its actual web search capabilities.
>
> 2. The benchmark features complex and challenging questions that typically require multiple steps of search, browsing, and reasoning, allowing for the evaluation of LLMs' agentic search abilities.
>
> 3. Unlike FreshQA and RealtimeQA, which rely on manually crafted Q&A pairs, LiveNewsBench includes an automated pipeline for data acquisition and Q&A generation. This allows for scalable dataset construction and includes a training set. For additional reliability, each benchmark release also includes a human-verified test set.
>
> Further, we would like to provide an **extended comparison with FreshQA**. LiveNewsBench features regularly-refreshed question-answer pairs that are grounded in recent news events. In contrast, **FreshQA features fixed questions that are not grounded in recent events**, and only the answers would change over time. The static nature and relatively simple design of FreshQA questions allow SotA LLMs to answer a large percentage of questions correctly via memorization  (Page 9, Lines 470-476). This limits FreshQA's effectiveness in evaluating the true search capabilities of LLMs. Furthermore, all FreshQA questions are manually crafted rather than automatically generated, which limits the scalability of the approach and results in the absence of a training set.
>
> To further illustrate the difference, we conducted additional experiments evaluating SotA LLMs on the FreshQA benchmark in an offline setting, on both the full dataset and the fast-changing subset. Our results confirm that current SotA LLMs perform well on FreshQA without search or Internet access, even on the fast-changing subset. Most notably, **Grok 4 achieves 74.6% accuracy on the full FreshQA dataset and 60.3% on the fast-changing subset without any internet access.** These results demonstrate that FreshQA can already be largely solved by existing models without using web search, and a regularly refreshed and factual QA benchmark is necessary for evaluating the search capabilities of LLMs.
>
> | Model     | Accuracy on LiveNewsBench without Internet | FreshQA Overall Accuracy | FreshQA Fast-Changing Subset Accuracy |
> |----------|-----------|------------|---------|
> | Grok 4                              | 23.5%                       | 74.6%            | 60.3%                  |
> | Qwen3 235B A22B Instruct 2507       | 12.5%                       | 65.4%            | 48.1%                  |
> | Qwen3 235B A22B Thinking 2507       | 14.5%                       | 68.0%            | 45.8%                  |
> | GPT-4.1                             | 18.5%                       | 68.0%            | 45.0%                  |
> | DeepSeek V3.1                       | 16.0%                       | 65.2%            | 42.0%                  |
> | GPT-5                             | 25.0%                       | 72.4%            | 40.5%                  |
> | Claude Sonnet 4                    | 12.5%                       | 65.4%            | 38.9%                  |
> | DeepSeek V3.1 Thinking              | 17.5%                       | 66.2%            | 38.2%                  |
>
> We use LLM-as-a-judge on the short-form final answers, which is most similar to the "relaxed" evaluation setting proposed by the FreshQA authors. To validate our setup, we also re-run the GPT-4 in our evaluation, and GPT-4 scored 39.0% in overall accuracy, and 20.6% in fast-changing subset, in line with the numbers reported in the original authors in the “relaxed” settings (46.4% and 14.4%, respectively).

---

> ### Author Response · Authors · 2025-11-21
> **Response to Reviewer F4vC, Continued**
>
> **In comparison to RealTimeQA**, we would like to highlight several differences of our benchmark that make it more suitable for evaluating web search capabilities of LLMs:
>
> 1. RealTimeQA is entirely based on crawling existing human-authored questions from news sources such as CNN and USA Today, rather than generating new, targeted questions. As such,  searching the given questions verbatim may lead to the source web page and leak the answer. Therefore it is not suitable for evaluating multi-step web search.
>
> 2. Each RealTimeQA release is significantly smaller, containing only 30 questions, compared to 200 in each LiveNewsBench human-verified test set and 800 in the full test set.
>
> 3. RealTimeQA does not provide ground-truth answers, but only "evidence" snippets, making it difficult to verify model responses. In many cases, the evidence field is missing, leaving the sample with no references.
>
> 4. The dataset has not been updated since January 2024, which predates the training data cutoff of most current SotA models.
>
> > Weakness 3: The human validation is done by an author of the paper, and the paper states that the human validator rejects about 20% of the automatically generated QA pairs. Since the authors could be biased, an additional validation should be done by independent human validators to get an unbiased estimate of QA quality and ideally to measure inter-human agreement as well.
>
> Thank you for raising this concern. We would like to clarify that our human review process is primarily conducted to enhance reliability and out of an abundance of caution. The ~20% of rejected questions may be slightly ambiguous or have answers that could change over time. However, they are often still answerable via search and do not significantly affect the overall benchmark results. As discussed in Section 4.2 (Page 7, 344-358) and Table 2 (page 6, 290-302, reproduced below), both the accuracy and the relative ordering of models on the full test set remain stable with those on the human-verified subset. This suggests that the fully automated test set is a reliable measure of LLM search capabilities.
>
> Additionally, in response to the feedback during the rebuttal, we are committed to further improving our QA quality evaluation via human annotation. We have already initiated a human evaluation study, which includes assessments of QA quality and measurements of inter-annotator agreement. We will report the results of this additional validation and update the paper PDF accordingly once the study is complete.
>
> | Model                         | Human-verified accuracy (%) | Full test accuracy (%) | Delta | Rank change (Human Verified → Full) |
> |-------------------------------|-----------------------------|------------------------|-----------:|----------------------------|
> | GPT-5                       | 91.0                        | 87.9                   |    -3.1    | 1 → 1                      |
> | DeepSeek V3.1 Thinking (685B) | 83.0                        | 84.4                   |    +1.4    | 3 → 2                      |
> | Claude Sonnet 4             | 83.5                        | 81.9                   |    -1.6    | 2 → 3                      |
> | Qwen3 235B A22B Instruct 2507 | 76.0                        | 76.2                   |    +0.2    | 4 → 4                      |
> | Gemini 2.5 Pro                | 75.5                        | 75.6                   |    +0.1    | 5 → 5                      |
>
>
> > Weakness 4: The benchmark questions test both search and reasoning capabilities. This can be seen as a strength, but on the other hand, the fact that these skills are mixed in a single question means that we cannot separate a model's basic reasoning ability (such as mathematical skills) from its ability to search for new knowledge.
>
> Thank you for raising this concern. We would like to confirm that our web search agent is implemented using a reflect-before-act approach, similar to ReAct [1]. Since reasoning also contributes to more effective search planning and keyword selection, this paradigm has become the default in many agentic LLM search systems. This includes both commercial and open-source web search agent implementations (e.g., OpenAI Deep Research [2], Claude Web Search [3], Open Deep Research [4], Kimi Researcher [5]). Therefore, we believe that including reasoning, as opposed to explicitly disallowing it, is a fair and realistic way to evaluate the agentic web search capabilities of LLMs.
>
> Moreover, many state-of-the-art models, such as GPT-5, Grok 4, and Gemini 2.5 Pro, always perform chain-of-thought (CoT) reasoning before producing responses. It is not possible to disable such CoT reasoning through API or prompting, and the content of the CoT is not returned to the user. As such, for any benchmark aiming to evaluate these models, it is inherently impossible to isolate the contribution of reasoning from other capabilities required for the task.

---

> ### Author Response · Authors · 2025-11-21
> **Response to Reviewer F4vC, Continued**
>
> > Weakness 5: A more minor weakness: The paper states that the benchmark will be updated on a quarterly basis. This may be sufficient for many purposes, but it does mean that when a new model comes out, it may take a full quarter before it can be meaningfully evaluated.
>
> We would like to clarify that it is not necessary to delay the evaluation of newly released models until the next benchmark update. This is because the training data cutoffs of state-of-the-art models are typically publicly disclosed, and they usually precede the model release date by 9-12 months[6,7]. Since our test set contains only events from the most recent two months at the time of each benchmark release, and the benchmark is refreshed quarterly, there is no temporal overlap with the models’ training data. Therefore, new models can be evaluated at any time without compromising the integrity of the benchmark.
>
> > Weakness 6: Also minor: The benchmark relies on GPT-4.1 and GPT-5 for question generation and evaluation. These are closed models whose availability and features are out of the authors' control, which could introduce instability into the benchmark updates.
>
> We would like to clarify that LiveNewsBench evaluation results are intended to be compared within the same release, not across different releases, as each release contains a distinct set of questions. This design is consistent with other regularly updated LLM benchmarks, such as LiveBench[8] and LiveCodeBench[9]. Within a given release, all Q&A pairs are generated and evaluated using the same set of LLMs, ensuring that evaluation results are comparable.
>
> In response to the feedback received during the rebuttal, we are currently working on implementing a version of LiveNewsBench using open-source LLMs, and will compare its evaluation results against the standard version built with proprietary models. This process requires re-running the benchmark and will therefore take additional time. We will update the comments and the paper accordingly once this is completed. Furthermore, while open-weight LLMs generally lag behind proprietary ones in performance, their capabilities are rapidly improving. As such, we expect that a version of LiveNewsBench built on open models will soon approach the quality of the version based on proprietary models available today.
>
> > Weakness 7: The provided URL, livenewsbench.com, does not appear to be live.
>
> Thank you for raising this issue. The website is now fully functional with links to the code, dataset, as well as the model output logs. We regret this error.
>
> > Question 1: I have a hard time following the longitudinal example in Fig. 2. What makes it longitudinal?
>
> Thank you for your detailed questions. In this example, “number of Ukrainian prisoners still
> remained to be released” may change over time, so it is considered a longitudinal question. We do not include longitudinal questions for our training, validation, and test sets due to the difficulty to produce an up-to-date ground truth answers, but researchers may use these questions for different purposes, such as measuring how responses to the longitudinal questions shift based on the varying knowledge-cutoff dates of LLMs.
>
> > Question 2: Table 3 doesn't appear to be referenced anywhere. I assume it should have been referenced in Sec. 4.3?
>
> Thank you for your detailed review. Yes, Table 3 was meant to be referenced in Section 4.3. We will correct the reference in the updated paper.
>
> Please let us know if you have any further questions or concerns. We are happy to address any additional issues you may have. Thank you again for your thoughtful feedback.
>
> [1]: ReAct: Synergizing Reasoning and Acting in Language Models
>
> [2]: Introducing deep research, OpenAI Blog
>
> [3]: How we built our multi-agent research system, Engineering at Anthropic Blog
>
> [4]: langchain-ai/open_deep_research, GitHub Repository
>
> [5]: Kimi-Researcher: End-to-End RL Training for Emerging Agentic Capabilities, Moonshot AI Blog
>
> [6]: Compare models, OpenAI Platform Documentation
>
> [7]: Anthropic’s Transparency Hub
>
> [8]: LiveBench: A Challenging, Contamination-Limited LLM Benchmark
>
> [9]: LiveCodeBench: Holistic and Contamination Free Evaluation of Large Language Models for Code

---

### Author Response · Authors · 2025-12-03
**Summary of Reviews and Author Responses**

First, we would like to thank the reviewers for reviewing our submission. We sincerely appreciate the time and constructive feedback. We would like to summarize the reviews and our rebuttals below.

- [Positive Reviewer Responses](#positive-reviewer-responses)
- [General Reviewer Concern: Comparisons to Prior Time-Sensitive Factual QA Benchmarks](#general-reviewer-concern-comparisons-to-prior-time-sensitive-factual-qa-benchmarks)
- [Other Concerns](https://openreview.net/forum?id=5HJkrZTtqr&noteId=auUz1clc8O)

# Positive Reviewer Responses
1. The motivation of the paper is clear, and reviewers agree that there is a need to better evaluate search-enabled models on events beyond their training to reduce data contamination (Reviewers F4vC, ri14).
2. The automated data curation and QA pair generation pipeline is reasonable and scalable, and the setup is likely to produce high-quality QA pairs (Reviewers F4vC, ByQq, ri14, HD95).
3. The inclusion of a human-verified test set strengthens benchmark robustness (Reviewers ByQq, HD95).
4. The benchmark is effective at differentiating model capabilities, as well as between online (search-enabled) and offline settings (Reviewers F4vC, ByQq, ri14, HD95).
5. The paper is well-written and easy to follow (Reviewers F4vC, ByQq).

# General Reviewer Concern: Comparisons to Prior Time-Sensitive Factual QA Benchmarks
Several reviewers (F4vC, ByQq, ri14) questioned the need for a new benchmark and how LiveNewsBench differs from FreshQA, Daily Oracle, and RealTimeQA. This was a central concern, and we highlight key evidence from our rebuttal:

1. **Earlier benchmarks can often be solved by LLMs’ internal knowledge alone.** Their questions tend to be simple and predictable, making it difficult to disentangle search capabilities from knowledge. To demonstrate this, we conducted **additional offline evaluations** on FreshQA and Daily Oracle. On **FreshQA**, Grok 4 achieves **74.6% accuracy on the full dataset and 60.3% on the fast-changing subset** without internet access. On **Daily Oracle**: GPT-5 achieves **77.0% accuracy** offline. By contrast, the best-performing model (GPT-5) **reaches only 25.0% accuracy on LiveNewsBench** offline, indicating that our benchmark truly requires effective search.

2. **LiveNewsBench is continuously refreshed and contains complex, multi-step questions.**
All Q&A pairs are regenerated every three months for at least two years, based on recent news events. These questions typically demand multi-hop search and reasoning.
In comparison, **FreshQA questions are static**, making it easier for models to rely on memorized knowledge. RealTimeQA does not release ground-truth answers and has not been updated since January 2024.

3. Unlike FreshQA and RealTimeQA, which rely on human-written questions, **LiveNewsBench uses an automated pipeline for Q&A pair generation**, substantially reducing maintenance overhead and enabling construction of a training set. While human verification is used for the human-verified subset, it is not essential for operating the benchmark.

---

> ### Author Response · Authors · 2025-12-04
> **Summary of Reviews and Author Responses, Continued**
>
> # Other Concerns
> ## Quality of Automatically Generated QA Pairs and Human Verification
>
> Reviewers (F4vC, ByQq, ri14, HD95) raised concerns about the ~20% rejection rate during human verification. We clarify that this verification step is intentionally conservative, filtering out questions that may be slightly ambiguous or whose answers could shift over time. These **rejected samples are often still answerable via search**.
>
> Benchmark stability is largely unaffected by the human verification process: across five evaluated models, accuracy differs by only 1.3% on average (maximum 3.1%, for GPT-5) between the full set and the human-verified subset (Table 2). This shows that the dataset is already of high quality prior to human review.
>
> ## Distinguishing Reasoning from Search Capability
>
> Reviewers (F4vC, ByQq, ri14) asked whether the ReAct-based evaluation conflates reasoning with search. Our agent uses a reflect-before-act paradigm, which is standard in modern agentic LLM systems (e.g., OpenAI Deep Research, Claude Web Search, Open Deep Research). Reasoning is integral to effective search planning (e.g., keyword selection, iterative search), and disabling it would be an unrealistic setting for evaluating LLM-based web search.
>
> Furthermore, state-of-the-art models such as GPT-5, Grok 4, and Gemini 2.5 Pro always perform chain-of-thought reasoning that cannot be disabled via API. Therefore, it is inherently impossible for any benchmark to perfectly isolate reasoning from search ability for such models.
>
> ## Dependence on Closed-Model APIs
>
> Reviewers (F4vC, ByQq) expressed concerns about potential instability due to reliance on closed-model APIs. LiveNewsBench is designed for within-release comparisons: each release contains a distinct set of questions, and all Q&A generation and evaluations are performed using the same models. Therefore, model API changes between releases would not affect evaluation. This within-release comparison aligns with other dynamically updated benchmarks such as LiveBench and LiveCodeBench.
>
> ## Long-Term Maintenance
>
> Reviewers (F4vC, ByQq, ri14, HD95) raised concerns about the benchmark’s stated two-year update period. The two-year window is a minimum commitment. We plan to continue updates contingent on community interest, particularly for the training, validation, and test sets that do not require human supervision. All code and data are fully open-sourced, enabling the community to extend or maintain the benchmark beyond our initial commitment.
>
>
> # Final Note
> While we did not get a response from the reviewers due to the shortened discussion period, we believe all concerns are addressed and no new issues were brought up.

---

### Meta-Review · Area_Chair_DTMD · 2026-01-06

**Summary:**

This paper present a new dynamic benchmark, aiming to measure the agentic searching ability of a model. Reviewers' concern on mainly one
- Novelty, it is conceptually similar to FreshQA and RealTimeQA
- Searching and reasoning are hard to disentagle
- Long-term value and uncertainty

**Reviewer Concerns:**

- Novelty, it is conceptually similar to FreshQA and RealTimeQA

Only partially addressed. While I agree there's some difference with previous work, I feel it is incremental, and the novelty falls below the bar of ICLR

- Searching and reasoning are hard to disentagle

Addressed, I agree with the authors that searching and reasoning are both part of agentic searching

- Long-term value and uncertainty

Not addressed but it is a fact for all dynamic benchmark and we need dynamic benchmark for search. However, I personally believe this project better fits a community project rather than an ICLR paper.

**Reviewer Scores:**

ByQq: 2 → 4

because new experiments and cost analysis were added

HD95: 2 → 4

because evaluation robustness concerns were addressed but long-term maintenance and data noise issue are still there

I believe other reviewers will keep the original rating.

---

### Decision · Program_Chairs · 2026-01-26

Reject